# Efficient Strike Artifact Reduction Based on 3D-Morphological Structure Operators from Filtered Back-Projection PET Images

**DOI:** 10.3390/s21217228

**Published:** 2021-10-30

**Authors:** Chun-Yi Chiu, Yung-Hui Huang, Wei-Chang Du, Chi-Yuan Wang, Huei-Yong Chen, Yun-Shiuan Shiu, Nan-Han Lu, Tai-Been Chen

**Affiliations:** 1Department of Radiology, Kaohsiung Veterans General Hospital, No. 386, Dazhong 1st Rd., Zuoying District, Kaohsiung City 81362, Taiwan; chiucys@vghks.gov.tw; 2Department of Information Engineering, I-Shou University, No. 1, Sec. 1, Xuecheng Rd., Dashu District, Kaohsiung City 84001, Taiwan; wcdu@isu.edu.tw; 3Department of Medical Imaging and Radiological Sciences, I-Shou University, No. 8, Yida Rd., Yanchao District, Kaohsiung City 82445, Taiwan; yhhuang@isu.edu.tw (Y.-H.H.); wang1b011@isu.edu.tw (C.-Y.W.); ed103911@edah.org.tw (N.-H.L.); 4Department of Nuclear Medicine, E-DA Hospital, No. 1, Yida Rd., Yanchao District, Kaohsiung City 82445, Taiwan; ed100397@edah.org.tw (H.-Y.C.); edps900002@edah.org.tw (Y.-S.S.); 5Department of Pharmacy, Tajen University, No. 20, Weixin Road, Yanpu Township, Pingtung City 90741, Taiwan; 6Department of Radiology, E-DA Hospital, I-Shou University, No. 1, Yida Road, Jiao-Su Village, Yanchao District, Kaohsiung City 82445, Taiwan

**Keywords:** PET, FBP, strike artifacts, MSO, ORC

## Abstract

Positron emission tomography (PET) can provide functional images and identify abnormal metabolic regions of the whole-body to effectively detect tumor presence and distribution. The filtered back-projection (FBP) algorithm is one of the most common images reconstruction methods. However, it will generate strike artifacts on the reconstructed image and affect the clinical diagnosis of lesions. Past studies have shown reduction in strike artifacts and improvement in quality of images by two-dimensional morphological structure operators (2D-MSO). The morphological structure method merely processes the noise distribution of 2D space and never considers the noise distribution of 3D space. This study was designed to develop three-dimensional-morphological structure operators (3D MSO) for nuclear medicine imaging and effectively eliminating strike artifacts without reducing image quality. A parallel operation was also used to calculate the minimum background standard deviation of the images for three-dimensional morphological structure operators with the optimal response curve (3D-MSO/ORC). As a result of Jaszczak phantom and rat verification, 3D-MSO/ORC showed better denoising performance and image quality than the 2D-MSO method. Thus, 3D MSO/ORC with a 3 × 3 × 3 mask can reduce noise efficiently and provide stability in FBP images.

## 1. Introduction

Positron emission tomography (PET) can provide functional images and identify abnormal metabolic regions of the whole body that effectively detect the presence and distribution of tumors. Because of the use of different radioactive tracers, factors associated with nuclear medicine imaging, including the physical characteristics of photons, data acquisition settings, or reconstruction algorithms that generate artifact noise in images, can easily interfere with the image quality [1,2,3,4,5,6,7]. Common noise models in PET scans include Gaussian, Poisson, and mixed noise. The generation of these noises will affect image capture, scan time, correction methods, and image reconstruction methods, thus affecting image quality. The filtered back-projection (FBP) algorithm is one of the most common methods for image reconstruction [8,9,10] and is characterized by the rapid collection of high-contrast reconstructed PET images with fewer photon counts and in a shorter time [11,12,13,14,15]. However, due to the limited number of projections in the algorithm, strike artifacts are easily produced [16], which affects the results of nuclear medical examinations and diagnoses.

Shih et al., who used 99mTC-TRODAT-1 to examine Parkinson disease (PD) in 1999, demonstrated the importance of effectively reducing background noise for clinical diagnosis [17]. Previous studies have shown that it is possible to remove strike artifacts using interpolation of projections by contouring, but it is challenging to select an interpolated value. For instance, methods such as the Brushlet, Wavelet, and Curvelet transformation can be used to remove strike artifacts from PET images; however, caution should be applied when selecting a threshold [18,19,20,21].

In 2014, Chen et al. used a morphological structure operation to effectively reduce strike artifacts in the FBP nuclear medicine image of the Deluxe Jaszczak phantom and employed the median Gaussian filter and various morphological structure operators (MSO; 2 × 2, 3 × 3, and 4 × 4) to reduce imaging noise [22]. Their findings clearly showed that the 3 × 3 MSO could produce images with a lower standard deviation of the image background, i.e., reducing image noise in the image background. According to the study by Chen et al., the use of MSO can provide noise removal of different angles and effectively remove background noise in each nuclear medicine image. However, during denoising, only a single image can be processed each time, and this process is quite time-consuming. Consequently, further investigations are needed to evaluate the feasibility of three-dimensional MSO.

Zhang et al. had published research about patch-based regularization and dictionary learning (DL) with computer simulation in 2019. The AD-based method was designed to balance between noise and quality of images. However, it is difficult to reduce noise and keep or improve quality of images due to encounter the low SNR of images or low count rates [23,24]. Meanwhile, Seo et al. proposed a new method, block sequential regularized expectation maximization (BSREM), to enhance the quality of images and accuracy of qualification as per occurred in low-count rates of PET scanning. In particular, the main arduous task is adjusting regularization parameter [25]. Moreover, Tatsumi et al. was mentioned regarding a Bayesian penalized likelihood method (BPL) to promote the gray levels of image as per low count rates condition. However, the small lesions usually affected the quantitative examination of PET images [26]. Moreover, the convolutional neural network (CNN) methods were incorporated with quantitative study under the conditions between low dose and sparse matrix on CT images. The experimental results were shown and were demonstrated to be able to improve the quality of noise images [27]. Regardless of the successfulness obtained by CNNs on CT images, the accuracy of quantitation in PET images is still to be confirmed [28,29].

This study was designed to use 2D/3D-MSO to eliminate strike artifacts in FBP-reconstructed PET images. Previous studies have shown that FBP can reconstruct images with higher contrast and a lower photon count in a shorter time. However, noise such as strike artifacts can be easily introduced into the images during reconstruction due to inadequate projection angles. These artifacts may affect the accuracy of the clinical diagnosis. Therefore, this work was anticipated to establish a complete 2D/3D-MSO denoising method that effectively reduces noise while maintaining better image quality.

## 2. Materials and Methods

### 2.1. Materials

In this study, the Deluxe Jaszczak phantom and rats were used for tomography, and the reconstructed images were obtained using the FBP algorithm. Additionally, the reconstructed images of the Deluxe Jaszczak phantom and rats were used to evaluate the background denoising, image resolution, and image quality merits after noise processing.

After the Deluxe Jaszczak phantom was injected with 74 MBq of 18F-Fluorine, it was scanned continuously for 20 min using a Siemens Biograph 6 PET-CT with a field of 700 mm (Figure 1). The Deluxe Jaszczak phantom image sizes were 168 × 168 and 65 slices that the total pixel sizes were 168 × 168 × 65.

The rats (Figure 2) were scanned using MicroPET R4 with a field of 63 mm. The images of the Deluxe Jaszczak phantom and rats were reconstructed using the FBP algorithm. The rat image sizes were 256 × 256 and 63 slices, and the total pixel sizes were 256 × 256 × 63.

### 2.2. Tomography Equipment and Settings

The Siemens Biograph 6 PET-CT was used to scan the Deluxe Jaszczak phantom with the following settings: field of view (FOV) 700 mm, span size 11, maximum ring difference 27, attenuation correction by computed tomography (CT), single-scatter simulation algorithm in Siemens software, delayed time window count for random correction, and FBP combined with Gaussian filters for image reconstruction.

The settings used for MicroPET R4 for rat tomography were as follows: FOV 63 mm, span size 3, maximum ring difference 31, attenuation correction by point source, no scatter correction, random correction by delayed time window count, image reconstruction using FBP, the number of projections 84, and the number of transaxial angles 96.

### 2.3. Research Flowchart and Experimental Design

The flowchart of this study is shown in Figure 3. The two-dimensional axial imaging method was used to run the two-dimensional MSO in each FBP nuclear medicine image of the Deluxe Jaszczak phantom and to identify the operator combined with the lowest background standard deviation. After identifying five combinations with the lowest background standard deviations, we used these five MSO combinations to perform noise processing on two-dimensional FBP images, and the output data were averaged. Finally, the standard deviation (STD), full width at half maximum (FWHM), and signal-to-noise ratio (SNR) were used to evaluate denoising and image quality after noise processing.

In addition, when 3D-MSO was used to process the FBP nuclear medicine images with the Deluxe Jaszczak phantom, a large number of operator combinations (for instance, morphological structural operator combinations of 3 × 3 × 3 had 2^27^, i.e., 134,217,728, combinations) required an extremely long duration when the operation was performed for each image. Therefore, the optimal response curve (ORC) was used to reduce the number of large datasets to 200 or less, and then 3D-MSO was used to identify the operator combinations with the lowest background noise for the images. After distinguishing five combinations with the lowest background standard deviations, we used the five MSO combinations to perform noise processing, and the output data were averaged. Finally, STD, FWHM, and SNR were used to evaluate denoising and image quality after noise processing.

### 2.4. Morphological Structure Operation

The purpose of applying the morphological structure operation to nuclear medicine images reconstructed by the FBP algorithm was to remove strike artifacts. Simultaneously, a parallel operation was used to identify the best MSO combinations quickly. The formula is provided in Equations (1) and (2).
(1)$$A_{2D}=\left(a_{ij}\right)=\left\{\left(\begin{array}{cc} a_{11} & a_{12}\\ a_{21} & a_{22} \end{array}\right)\;or\;\left(\begin{array}{cc} a_{11} & a_{12}\\ a_{21} & a_{22}\\ a_{31} & a_{32} \end{array}\right)\;or\;\left(\begin{array}{ccc} a_{11} & a_{12} & a_{13}\\ a_{21} & a_{22} & a_{23} \end{array}\right)\;or\;\left(\begin{array}{ccc} a_{11} & a_{12} & a_{13}\\ a_{21} & a_{22} & a_{23}\\ a_{31} & a_{32} & a_{33} \end{array}\right)\right\}$$
(2)$$A_{3D}=\left(a_{ij}\right)=\;\left\{\left(\begin{array}{cc} a_{11} & a_{12}\\ a_{21} & a_{22} \end{array}\right),\;\left(\begin{array}{cc} a_{11} & a_{12}\\ a_{21} & a_{22} \end{array}\right)\;or\ldots or\;\left(\begin{array}{ccc} a_{11} & a_{12} & a_{13}\\ a_{21} & a_{22} & a_{23}\\ a_{31} & a_{32} & a_{33} \end{array}\right),\;\left(\begin{array}{ccc} a_{11} & a_{12} & a_{13}\\ a_{21} & a_{22} & a_{23}\\ a_{31} & a_{32} & a_{33} \end{array}\right),\;\left(\begin{array}{ccc} a_{11} & a_{12} & a_{13}\\ a_{21} & a_{22} & a_{23}\\ a_{31} & a_{32} & a_{33} \end{array}\right)\right\}$$
where *a_ij_* is a value between 0 or 1 in column *i* at row *j*, where 0 represents the denoising structure, and 1 represents the structure of the reserved signals. The sizes of the two-dimensional morphological structures used in this study were 2 × 2, 2 × 3, 3 × 2, and 3 × 3, and the number of operator combinations were 24 (16), 26 (64), 26 (64), and 29 (512). The sizes of the three-dimensional MSOs used were 2 × 2 × 2, 2 × 3 × 2, 3 × 2 × 2, 3 × 3 × 2, 2 × 2 × 3, 2 × 3 × 3, 3 × 2 × 3, and 3 × 3 × 3. The number of operator combinations were 2^8^ (256), 2^12^ (4096), 2^12^ (4096), 2^18^ (262,144), 2^12^ (4096), 2^18^ (262,144), 2^18^ (262,144), and 2^27^ (134,217,728). The principle of matrix convolution was used to perform the operation in the morphological structure matrix. Convolution is a mathematical operation that produces a third function by using two functions, *f*(*t*) and *g*(*t*), i.e., it represents the product of the function *f*(*t*) and the flipping and translating function *g*(*t*). The area enclosed by the curve is defined as Equation (3).
(3)$$(f\times g)(t)\overset{\mathrm{def}}{=}\int_{R^{n}}^{}f(x)\;g(t-x)dx$$

Functions *f*(*t*) and *g*(*t*) can be measured on *R^n^*, *t* ∈ (−∞, ∞). When *f*(*t*) is convoluted with *g*(*t*), it is denoted as *f* × *g*, which represents the integral of the product of a flipping and translating function with another function. Therefore, *X^input^* is an input of the FBP nuclear medicine image and *B_m_* is an image obtained by an *A_m_* morphological structure matrix operation, as shown in Equation (4), where m is the number of combinations.
(4)$$B_{m}=X^{input}\cdot A_{m},\;m=1,\;2,\;\ldots ,\;2^{L}$$

The two-dimensional and three-dimensional morphological matrix operations on FBP nuclear medicine images were performed, and the image background denoising merits, image quality, and operation time were compared. The three-dimensional MSO with more de-structured phases was expected to provide a quality image with reduced background noise (i.e., strike artifacts) compared with the two-dimensional MSO.

### 2.5. Optimal Response Curve

This method is based on the concept of an optimal smooth response function. Several functions are similar and correspond to the best response curves (Figure 4). Many functions of the smooth best response curve can be calculated by the interpolation method to obtain the best standard response curve. The definition of response curve is shown as Equation (5).
(5)Response Curve=eE(xi)γeE(xi)γ+eE(xi−1)γ, i=1,2,3,4.
where *E*(*x*) is the predicted response behavior of *x*, and *γ* is a parameter value of the function that deviates from the parameter value of the true best response function. A larger *γ* value also indicates a larger deviation from the actual best response function.

Therefore, peaks exist in the curve of the exponential function and quadratic function, and the definition can be used to calculate the best result. The number of MSO combinations is an exponential function (such as 2^9^ and 2^27^), and therefore, the optimal response curve is used to reduce the number of combinations during the operation and identify the contributing operator combination. As shown in Figure 4, the data for a curve are assumed to be large and ranges from X_1_ to X_4_. The sequential calculation is time-consuming. Therefore, the best response curve can be used to reduce the amount of and effectively screen the data. One hundred data points were selected between X_1_ and X_4_, with each point corresponding to a functional value representing the image background standard deviation. The background standard deviation was calculated for each of the 100 data points to determine the two data points with the lowest values, namely, X_2_ and X_3_. By repeating this method, the background standard deviation was calculated for each of the 100 data points between X_2_ and X_3_ to identify another two data points with the lowest values. After the method was repeated several times, the amount of data was reduced to 200 pairs, which is the minimum value required for the study, and the morphological structure operation was then performed.

The morphological structure operation and optimal response curve can effectively identify contributing data and reduce the computation time during the process. Thus, the time-consuming 3D-MSO can remove the background noise on FBP nuclear medicine images over a shorter processing time and simultaneously improve the image quality.

### 2.6. Image Background Value

The noise assessment was performed by using the region of interest (ROI) to select images of the Deluxe Jaszczak phantom and rat (Figure 5) and calculate their background standard deviation to assess image background denoising merits after MSO noise processing. A smaller standard deviation of the background in operation was associated with less background noise, indicating a better MSO denoising effect. The standard deviation of the image background was calculated as shown in Figure 5 (yellow region).
(6)$$N_{B}=\sum_{x}^{}\sum_{y}^{}\delta \left(x,\;y\right),\;\delta \left(x,\;y\right)=\{\begin{array}{c} 1,\;if\;\left(x,\;y\right)\;\;outside\;of\;ROI\;\\ 0,\;elsewhere \end{array}$$
(7)$$\mu_{m\_B}=\frac{1}{N_{B}}\overset{}{\underset{x}{\sum }}\overset{}{\underset{y}{\sum }}\overset{}{\underset{i}{\sum }}\overset{}{\underset{j}{\sum }}A_{m}\left(i,\;j\right)I\left(x-i,\;y-j\right)$$
(8)$$STD_{m\_B}=\sqrt{\frac{1}{N_{B}}\sum_{x}^{}\sum_{y}^{}\sum_{i}^{}\sum_{j}^{}{\left[A_{m}\left(i,\;j\right)I\left(x-i,\;y-j\right)-\mu_{m\_B}V\right]}^{2}}$$
(9)$$Mean\;STD_{m\_B}=\frac{1}{S}\sum_{1}^{S}\sqrt{\frac{1}{N_{B}}\sum_{x}^{}\sum_{y}^{}\sum_{i}^{}\sum_{j}^{}{\left[A_{m}\left(i,\;j\right)I\left(x-i,\;y-j\right)-\mu_{m\_B}\right]}^{2}}$$
where *N_B_* is the total number of background pixels outside the ROI in the NxN image, as shown in Equation (6); *δ*(*x*, *y*) is the position of the pixel in the image; *I*(*x* − *i*, *y* − *j*) is an image background pixel value; *Am*(*i*, *j*) is the average pixel value of the MSO-processed image background in the nxn morphological structure matrix and *μm_B*, as shown in Equation (7); *STDm_B* is the image background standard deviation after MSO processing, as shown in Equation (8); *Mean STDm_B* is the mean standard deviation of the image background after MSO processing; and S is the number of scan slices, as shown in Equation (9).

### 2.7. Signal-to-Noise Ratio (SNR)

The ROI was used to select the Deluxe Jaszczak phantom and Harrington glands, sublingual glands, and brain of rats (Figure 6), and the SNRs of the ROIs were calculated to evaluate the image quality merits after MSO processing. A more substantial SNR value during the calculation was associated with a larger signal-noise-ratio, indicating better signals and less noise after MSO processing. The SNR formula is as follows (Equations (10)–(13)):
(10)$$N_{R}=\sum_{x}^{}\sum_{y}^{}\delta \left(x,\;y\right),\;\delta \left(x,\;y\right)=\{\begin{array}{c} 1,\;if\;\left(x,\;y\right)\;\;inside\;of\;ROI\\ 0,\;elsewhere \end{array}$$
(11)$$\mu_{m\_R}=\frac{1}{N_{R}}\sum_{x}^{}\sum_{y}^{}\sum_{i}^{}\sum_{j}^{}A_{m}\left(i,\;j\right)I\left(x-i,\;y-j\right)$$
(12)$$STD_{m\_R}=\sqrt{\frac{1}{N_{R}}\sum_{x}^{}\sum_{y}^{}\sum_{i}^{}\sum_{j}^{}{\left[A_{m}\left(i,\;j\right)I\left(x-i,\;y-j\right)-\mu_{m\_R}\right]}^{2}}$$
(13)$$SNR_{R}=\frac{\mu_{m\_R}}{STD_{m\_R}}$$
where *N_R_* is the total number of ROI pixels in the N × N image, as shown in Equation (10); *δ*(*x*, *y*) is the position of the pixel in the image; *I*(*x* − *i*, *y* − *j*) is the pixel value of the image ROI; *Am*(*i*, *j*) is the average pixel value of the MSO-processed image background in the N × N morphological structure matrix and *μm_R*, as shown in Equation (11); *STDm_R* is the standard deviation of the ROI image after MSO processing, as shown in Equation (12); and *SNR_R_* is the signal-to-noise ratio of the ROI image after MSO processing, as shown in Equation (13).

### 2.8. Image Resolution 

Image resolution refers to the ability of a display system or measurement method to distinguish details. This concept applies to time and spatial fields. The commonly used resolution is used for image clarity, and a higher resolution indicates that more detail can be visualized, i.e., better image quality can be obtained. The FWHM (full width at half maximum) is often used to assess the degree of image resolution.

This study used this method to calculate the FWHM of the Deluxe Jaszczak phantom image (Figure 7a–d). Total aperture of line-profiles was obtained individually and the FWHM was calculated. The purpose of this method was to delineate a line profile for the largest diameter aperture, i.e., the line profile located close to the maximum aperture diameter of the original object, in order to evaluate the image resolution merits after MSO processing.

Six line-profiles in the Deluxe Jaszczak phantom image were selected to estimate the FWHM in comparison with the designed diameter (Figure 7e–j), which were defined from Line1, Line2, Line3, Line4, Line5, and Line6. The FWHM was evaluated to evaluate and indicate the resolution after processing by MSO. Meanwhile, the intensities between raw and after processing by MSO showed similarity.

### 2.9. Computation Time Calculation

In this study, a unified and specific computer was used for computation time calculation of MSO processing, as shown in Table 1. In addition, the optimal response curve of the 3 × 3 MSO was used to reduce the number of combinations before performing MSO processing. In addition to comparing the differences in MSO computation time when ORC was not used, the ORC application’s feasibility is also discussed.

## 3. Results

In this study, 2D-and 3D-MSO were used to denoise FBP images of the Deluxe Jaszczak phantom and rats. The lowest background standard deviations were obtained from 3 × 3 ORC (2D) and 3 × 3 × 3 (3D) MSO, suggesting optimal background denoising merits.

All background standard deviation results were obtained by 2D- and 3D-MSO and are shown in Table 2. Simultaneously, compared with background noise of raw data, the Deluxe Jaszczak phantom and rat of the two-dimensional 3 × 3 MSO background noise were reduced by approximately 85.3% and 33.2%, respectively; the Deluxe Jaszczak phantom and rats of the three-dimensional 3 × 3 × 3 MSO background noise were reduced by approximately 87.1% and 55.3%, respectively. The results showed that both two-dimensional and three-dimensional MSO processing were effective in removing image background noise as per measurement by background standard deviations.

The SNR (signal to noise ratio) and CR (contrast ratio) were used to evaluate the image quality and contrast after MSO processing. Compared to the SNR of raw data without MSO processing, the SNR of the Deluxe Jaszczak phantom and the sublingual gland of the rat increased by approximately 27.2% and 8.2%, respectively, after two-dimensional 3 × 3 MSO processing, and approximately 28.00% and 12.00%, after three-dimensional 3 × 3 × 3 MSO processing. All SNR results were obtained by 2D- and 3D-MSO and are shown in Table 3. Moreover, compared to the CR of raw data without MSO processing, the SNR of the Deluxe Jaszczak phantom and the sublingual gland of the rat increased by approximately 30.02% and 4.62%, respectively, after two-dimensional 3 × 3 MSO processing, and approximately 121.93% and 87.94%, respectively, after three-dimensional 3 × 3 × 3 MSO processing. All SNR results were obtained by 2D- and 3D-MSO and are shown in Table 3.

The FWHM of the six cold spots in Deluxe Jaszczak phantom was used to evaluate the image resolution (Figure 7). The estimated FWHM of six spots are listed in Table 4. After 2D MSO processing, the FWHM were 33.33, 25.00, 16.67, 12.50, 12.50, and 8.33 mm. Simultaneously, after 3D MSO processing, the FWHM were 29.17, 25.00, 18.75, 12.50, 12.50, and 10.42. The estimated FWHM generated by 2D or 3D MSO was close to the designed diameters in the Deluxe Jaszczak phantom.

When 3D-MSO was used for the denoising process, numerous operator combinations required an extremely long duration and robust computing equipment when the operation was performed for each image. The 3D-MSO with ORC was used to effectively decrease computing time and load for general computing machine. The computing configurations for calculating MSO noise processing time is shown in Table 1, and use of MSO denoising processing of operation time is shown Table 5. The computational time of MSO processing was calculated to evaluate the overall benefits. The computational times for the two-dimensional MSO processing of the Deluxe Jaszczak phantom and rat images were 1925.38 and 3445.90 s, respectively, while those for three-dimensional MSO processing with ORC were 219.09 and 100.08 s, respectively. The results revealed a shorter computational time for the three-dimensional MSO. This difference can be explained by using the ORC first to reduce the number of combinations followed by MSO processing. Moreover, 29 combinations and one-by-one noise processing resulted in more prolonged time consumption with the two-dimensional MSO. Moreover, after the ORC was used to reduce the number of operator combinations in the two-dimensional MSO, the computational time was clearly reduced from 1925.38 and 3445.90 s to 603.81 and 953.08 s, respectively. However, the duration was still longer than that observed for the three-dimensional MSO. This difference is embedded in the slice-by-slice processing required for the two-dimensional MSO (Figure 8).

The results showed that FBP images of the Deluxe Jaszczak phantom and rats processed with the three-dimensional MSO had lower image background noise in the visualization (Figure 9). Meanwhile, the rat images were confirmed without loss medical information by three senior independent radiologists in hospital.

## 4. Discussion

The most significant advantage of 3D-MSO is to preserve most diagnostic imaging information and effectively remove noise from nuclear medicine images. Although MSO has provided structural operators to remove image noise, such as linear, diamond, circular, square, and custom geometry, the strike artifacts on FBP images exhibit a 360° radiated shape in the 2D image and a 3D radiated shape in the 3D image. The strike artifacts’ density is related to the center; therefore, a single MSO geometric structure operator cannot adequately perform denoising. A variety of MSO geometric structure operators are required to achieve denoising. A significant challenge is the time-consuming process of identifying a reasonable and practical geometric structure operator (i.e., the evaluation may require days to weeks of CPU operation time). A possible solution may be to combine the ORC and parallel arithmetic processing modes and use limited sampling to identify reasonable operator combinations, quickly determining a reasonable and adequate geometric structure operator and reducing the CPU operation time.

Table 6 shows the comparisons among published literatures with the presented methods. Recently, the deep learning methods have been popularly applied to medical and molecular images to reduce noise [24,25] and enhance quality of images [26,27,28,29,30,31,32] including deep learning methods and generative adversarial networks algorithms (GAN). Meanwhile, the classical methods include patch-based regularization algorithms (PBRA) [24,25] and Bayesian penalized likelihood reconstruction algorithm (BSREMA) [26], and 2D and 3D MSO with ORC are applied to perform noise reduction with acceptable results.

This study strived to combine the advantages of 2D/3D MSO and ORC to develop a new algorithm that can effectively process FBP strike artifacts to improve the quality and resolution of FBP PET images and maintain image contrast. After noise processing, the image exhibits low noise in the iterative image and further preserves the high contrast advantage of the FBP image. In the future, PET imaging with a low count rate can be used to provide a better quality of the reconstructed image. The 2D and 3D MSO operation might be useful in the reduction of strike artifact in the FBP images.

## 5. Conclusions

In this study, 3D-MSO was used to perform noise processing on FBP nuclear medicine images to effectively reduce background noise and improve image quality. Moreover, combined with the ORC, 3D-MSO could effectively select contributing data and significantly reduce the computational time without compromising image quality. The contrast and quality of images were improvement by evaluated the SNR and CR. Meanwhile, according to the investigation of FWHM of cold spots in the phantom, the geometrical properties were preserved after processing by MSO. 

## Figures and Tables

**Figure 1 sensors-21-07228-f001:**
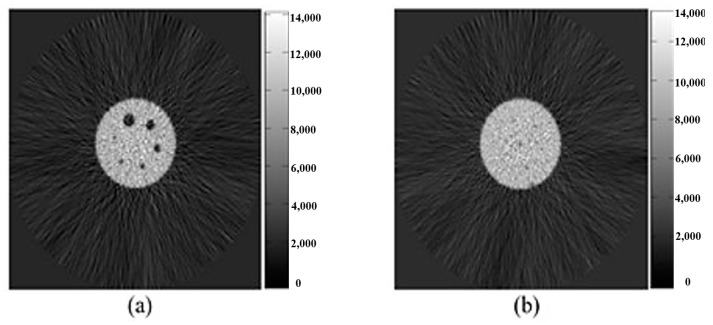
FBP images of the Deluxe Jaszczak phantom that was scanned continuously for 20 min with 74 MBq of ^18^F-Fluorine on the 1st (**a**) and 32nd (**b**) slices.

**Figure 2 sensors-21-07228-f002:**
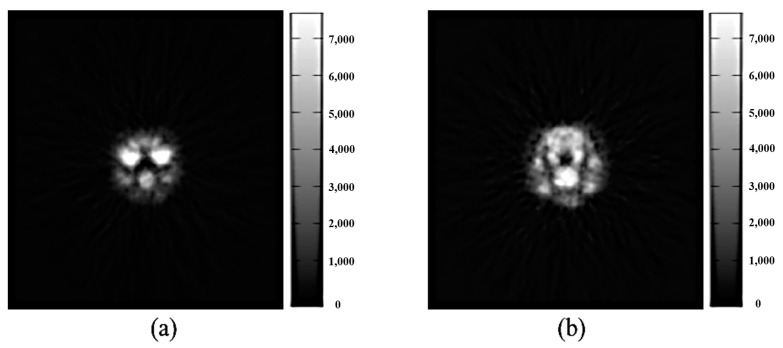
FBP images of the rats that were scanned using MicroPET R4 on the 1st (**a**) and 32nd (**b**) slices.

**Figure 3 sensors-21-07228-f003:**
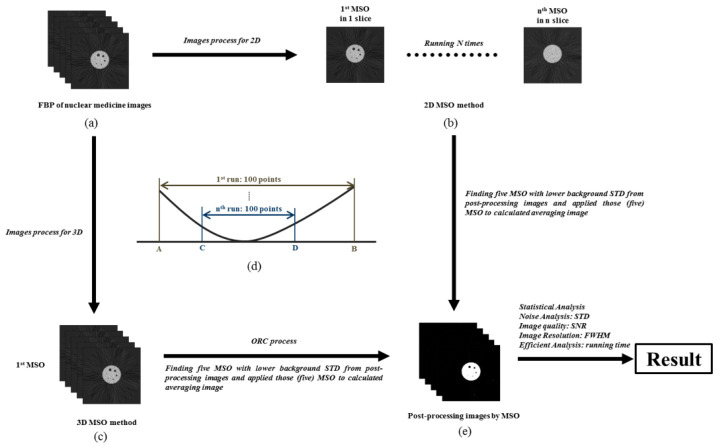
Study flow chart. The two-dimensional axial imaging method was used to run. 2D-MSO was used to process the FBP Deluxe Jaszczak phantom images with ORC. The two-dimensional axial nuclear medicine images (**a**). The images that were processed by 2D-MSO (**b**). The images that were processed by 3D-MSO (**c**). The operator data were defined and optimized by the principle of ORC (**d**). The post-processing images were quantified after three-dimensional MSO with ORC (**e**).

**Figure 4 sensors-21-07228-f004:**
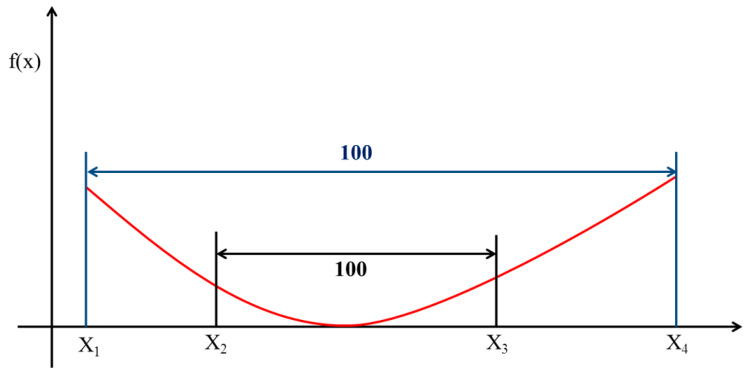
A diagram of the optimal response curve was calculated by the interpolation method and effectively identify contributing data and reduce the computation time during the process.

**Figure 5 sensors-21-07228-f005:**
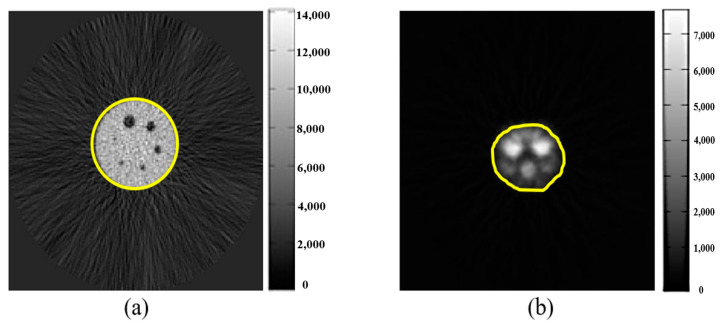
ROI method to determine the background of Deluxe Jaszczak phantom (**a**) and rat image (**b**), calculating their background standard deviation to assess image background denoising merits after MSO noise processing.

**Figure 6 sensors-21-07228-f006:**
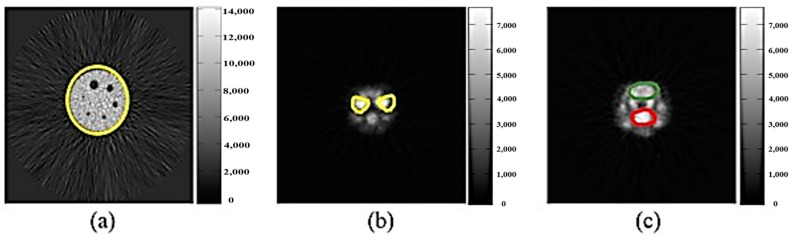
ROI selection of the (**a**) Deluxe Jaszczak phantom and (**b**) Harrington gland, sublingual gland, and (**c**) brain of rats; the SNRs of the ROIs were calculated to evaluate the image quality after MSO processing.

**Figure 7 sensors-21-07228-f007:**
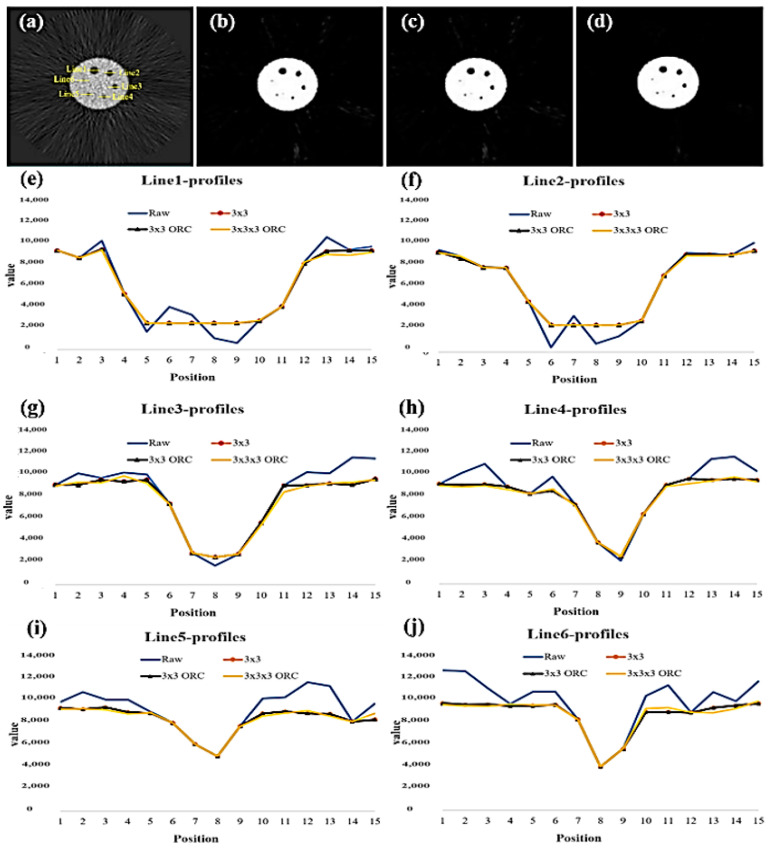
Total aperture of line-profiles in the FBP image of Deluxe Jaszczak phantom is shown (**a**), and FWHM were computed individually. A post-processing image by a 2D 3 × 3 MSO (**b**) and a 2D MSO with ORC (**c**). A post-processing image by a 3D 3 × 3 × 3 MSO with ORC (**d**). All aperture of line-profiles with/without MSO are shown (**e**–**j**).

**Figure 8 sensors-21-07228-f008:**
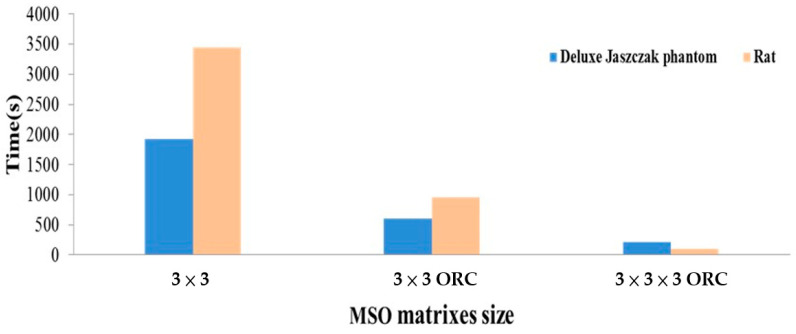
Evaluation of the computational time for MSO-processed FBP images of the Deluxe Jaszczak phantom and rats. The results revealed a shorter computational time by ORC.

**Figure 9 sensors-21-07228-f009:**
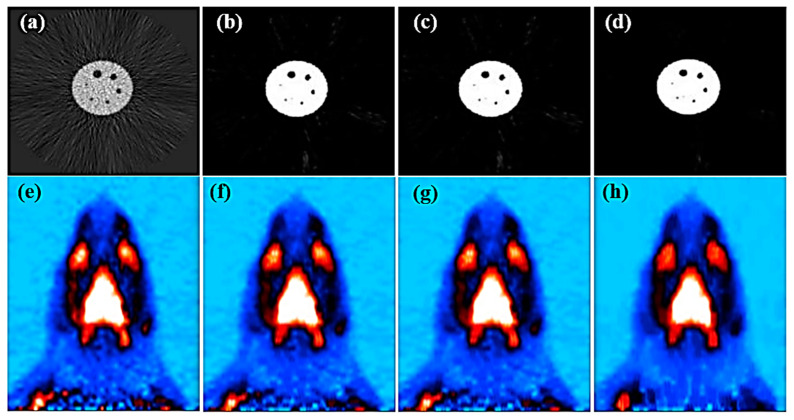
Demonstration the input images by MSO ORC operation on the FBP images for the Deluxe Jaszczak phantom (**a**) and a rat (**e**). A post-processing image by a 2D 3 × 3 MSO (**b**,**f**) and a 2D MSO with ORC (**c**,**g**). A post-processing image by a 3D 3 × 3 × 3 MSO with ORC (**d**,**h**). Notice that (**f**,**g**) were almost the same after 2D MSO with or without ORC in this study.

**Table 1 sensors-21-07228-t001:** The computing configurations for calculating MSO operation.

Item	Contents
Operating system	Windows 64-bit operating system
Central processing unit	Intel Core 2 Quad CPU Q6600
Computer memory	2.00 GB

**Table 2 sensors-21-07228-t002:** The Deluxe Jaszczak phantom and a rat of background standard deviations were obtained by 2D- and 3D-MSO with percentage of reducing noise. The percentage of reducing noise is defined as (Background STD in Raw − Background STD after MSO)/(Background STD in Raw) × 100%.

MSO Matrix Size	Background STD in Phantom	Background STD in Rat	Reducing Noise in Phantom	Reducing Noise in Rat
Raw ^1^	750.59	6.27	-	-
2 × 2	238.94	4.91	68.2%	21.7%
2 × 3	174.48	4.64	76.8%	26.0%
3 × 2	171.67	4.65	77.1%	25.8%
3 × 3	110.64	4.19	85.3%	33.2%
3 × 3 ORC ^2^	110.64	4.19	85.3%	33.2%
2 × 2 × 2 ORC	141.15	3.65	81.2%	41.8%
2 × 3 × 2 ORC	112.15	3.48	85.1%	44.5%
3 × 2 × 2 ORC	104.50	3.50	86.1%	44.2%
3 × 3 × 2 ORC	102.98	3.25	86.3%	48.2%
2 × 2 × 3 ORC	112.53	3.29	85.0%	47.5%
2 × 3 × 3 ORC	98.14	3.03	86.9%	51.7%
3 × 2 × 3 ORC	98.20	3.24	86.9%	48.3%
3 × 3 × 3 ORC	96.69	2.80	87.1%	55.3%

^1^ Raw data of FBP image without 2D- and 3D-MSO; ^2^ FBP image with 2D-MSO and ORC.

**Table 3 sensors-21-07228-t003:** The contrast ratio and signal-to-noise ratio were calculated for Deluxe Jaszczak phantom and rat images and were processed by 2D- and 3D-MSO. The percentage of increasing quality is defined as (SNR after MSO − SNR in Raw)/(SNR in Raw) × 100%. The CR is defined as (Mean intensity of Region in Phantom or rat)/(Mean intensity of background in Phantom or rat). The percentage of increasing contrast is defined as (CR in Region − CR in Background)/(CR in Background) × 100%.

MSO Matrix Size	CR/SNR in Phantom	CR/SNR in Rat	Increasing Contrast/Quality in Phantom	Increasing Contrast/Quality in Rat
Raw ^1^	263.44/4.53	174.64/14.21	-	-
2 × 2	66.94/5.19	157.81/14.53	−74.59%/14.6%	−9.64%/2.3%
2 × 3	110.81/5.41	165.53/15.00	−57.94%/19.4%	−5.22%/5.6%
3 × 2	113.30/5.41	165.25/14.81	−56.99%/19.4%	−5.38%/4.2%
3 × 3	342.52/5.76	182.70/15.38	30.02%/27.2%	4.62%/8.2%
3 × 3 ORC ^2^	342.52/5.76	182.70/15.38	30.02%/27.2%	4.62%/8.2%
2 × 2 × 2 ORC	179.32/5.51	209.00/15.39	−31.93%/21.6%	19.67%/8.3%
2 × 3 × 2 ORC	320.34/5.68	221.07/15.84	21.60%/25.4%	26.59%/11.5%
3 × 2 × 2 ORC	378.69/5.73	218.28/15.89	43.75%/26.5%	24.99%/11.8%
3 × 3 × 2 ORC	429.41/5.78	240.26/16.51	63.00%/27.6%	37.57%/16.2%
2 × 2 × 3 ORC	284.67/5.55	254.07/15.85	8.06%/22.5%	45.48%/11.5%
2 × 3 × 3 ORC	448.64/5.69	286.22/16.08	70.30%/25.6%	63.89%/13.2%
3 × 2 × 3 ORC	455.88/5.66	251.22/15.67	73.05%/24.9%	43.85%/10.3%
3 × 3 × 3 ORC	584.64/5.80	328.22/15.91	121.93%/28.0%	87.94%/12.0%

^1^ Raw data of FBP image without 2D- and 3D-MSO; ^2^ FBP image with 2D-MSO and ORC.

**Table 4 sensors-21-07228-t004:** The estimated FWHM was compared with reality.

FWHM of Line#	Aperture Size with/without MSO (mm)				
	Reality	Raw	3 × 3	3 × 3 ORC	3 × 3 × 3 ORC
Line1	31.80	29.17	33.33	33.33	29.17
Line2	25.40	20.83	25.00	25.00	25.00
Line3	19.10	14.58	16.67	16.67	18.75
Line4	15.90	12.50	12.50	12.50	12.50
Line5	12.70	10.42	12.50	12.50	12.50
Line6	9.50	10.42	8.33	8.33	10.42

**Table 5 sensors-21-07228-t005:** The computational time in seconds by 2D- and 3D-MSO operation.

MSO Matrix Size	Processing Time(s)	
	Deluxe Jaszczak Phantom	Rat
2 × 2	37.86	53.73
2 × 3	184.08	288.88
3 × 2	184.29	287.39
3 × 3	1925.38	3445.9
3 × 3 ORC ^1^	603.81	953.08
2 × 2 × 2 ORC	26.38	25.83
2 × 3 × 2 ORC	33.57	36.95
3 × 2 × 2 ORC	64.36	36.89
3 × 3 × 2 ORC	40.47	172.97
2 × 2 × 3 ORC	34.26	80.10
2 × 3 × 3 ORC	70.87	91.28
3 × 2 × 3 ORC	36.66	91.24
3 × 3 × 3 ORC	219.09	100.08

^1^ FBP image with 2D-MSO and ORC.

**Table 6 sensors-21-07228-t006:** The comparisons among published literature with the presented methods.

Authors	Year	Modality	Task	Method	Finding
Gao et al. [23,24]	2020	PET	Reduce noise	PBRA	It is difficult to reduce noise and keep or improve quality of images due to encounter the low SNR of images or low count rates
Seo et al. [25]	2020	PET	Improvement quality	BSREMA	To enhances the quality of images and accuracy of qualification as per occurred in low-count rates of PET scanning
Tatsumi et al. [26]	2021	PET/CT	Improvement quality of image	BPLRA	To promote the gray levels of image as per low count rates condition
Leuschner et al. [27]	2021	CT	Improvement quality of image	Deep learning methods	The experimental results were shown and demonstrated to be able to improve the quality of noise images
Yu et al. [28]	2020	Medical image synthesis	Reduce noise, enhance quality	Deep learning methods, 3D GAN	To promote the quality of image under low count rates or low dose
Soren et al. [29]	2020	CT, MRI, PET/MRI, PET/CT,	Reduce noise, enhance quality	GAN (GANs)	These novel models made a great impact on the computer vision field
Podgorsak et al. [30]	2021	CT	CT artifact correction	GANs	Improvement of reconstructed image quality under sparse angles
Koshino et al. [31]	2021	Medical and molecular imaging	Reduce noise, enhance quality	GANs	GANs are promising tools for medical and molecular imaging for promoted quality of images
Wang et al. [32]	2018	PET	Reduce noise, enhance quality	3D GANs	GANs are promising tools for improvement quality of image under low count rates
** *Presented Methods* **	2021	PET	Reduce noise, enhance quality	2D and 3D MSO with ORC	Demonstrated efficiently perform denoising processing and obtained acceptable quality of images

Notice: PBRA = patch-based regularization algorithm; BSREMA = block sequential regularized expectation maximization algorithm; BPLRA = Bayesian penalized likelihood reconstruction algorithm; GAN = generative adversarial networks.

## Data Availability

Not applicable.

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
