# Peer review of "Efficient Strike Artifact Reduction Based on 3D-Morphological Structure Operators from Filtered Back-Projection PET Images"

_sensors, 2021, doi:10.3390/s21217228_

Round 1
Reviewer 1 Report
In this manuscript, 2D and 3D morphological structure operators (MSO) were used to denoise positron emission tomography (PET) images after applying the filtered back-projection (FBP) algorithm. The goal is to eliminate strike artifacts without reducing image quality. The background standard deviation was used to evaluate the tested methods. Here are my comments and questions:
- In my opinion, the literature review on the artifacts and noise reduction methods should be more in-depth and insightful. Could you please elaborate on the fragment concerning the state-of-the-art method used in the literature?
- How many images were used in the experimental verification of your method? Could you support the conclusion that two-dimensional and three-dimensional MSO processing were effective in removing image background noise by including some quantitative analysis?
- How do you estimate the percentage of noise reduction?
- Could you provide some exemplary images showing how your method reduces the artifact?
Author Response
Reviewer 1
In this manuscript, 2D and 3D morphological structure operators (MSO) were used to denoise positron emission tomography (PET) images after applying the filtered back-projection (FBP) algorithm. The goal is to eliminate strike artifacts without reducing image quality. The background standard deviation was used to evaluate the tested methods. Here are my comments and questions:
- In my opinion, the literature review on the artifacts and noise reduction methods should be more in-depth and insightful. Could you please elaborate on the fragment concerning the state-of-the-art method used in the literature?
[Reply]: Thanks reviewer constructive comments and suggestions.
We had modified the Section of Introduction in this revision by updated some latest published literatures as below.
Zhang et al. had published a research about patch-based regularization and dictionary learning (DL) with computer simulation in 2019. The AD-based method was designed to balance between noise and quality of images. However, it is difficult to reduce noise and keep or improve quality of images due to encounter the low SNR of images or low count rates [24, 25]. Meanwhile, Seo et al. was proposed a new method, block sequential regularized expectation maximization (BSREM), to enhances the quality of images and accuracy of qualification as per occurred in low-count rates of PET scanning. Especially, the main arduous task is adjusting regularization parameter [26]. Also, Tatsumi et al. was mentioned about a Bayesian penalized likelihood method (BPL) to promote the gray levels of image as per low count rates condition. However, the small lesions were usually effected the quantitative examination of PET images [27]. Moreover, the convolutional neural network (CNN) methods were incorporated with quantitative study under the conditions between low dose and sparse matrix on CT images. The experimental results were shown and demonstrated to be able to improve the quality of noise images [28]. Although, the grateful successfulness obtained by CNNs on CT images, the accuracy of quantitation in PET images is still to be confirmation.
- How many images were used in the experimental verification of your method? Could you support the conclusion that two-dimensional and three-dimensional MSO processing were effective in removing image background noise by including some quantitative analysis?
[Reply]: Thanks reviewer constructive comments and suggestions.
The numbers of images between Deluxe Jaszczak phantom and rat are 65 and 63 slices, respectively. The volume matrix between Deluxe Jaszczak phantom and rat are 168×168×65 and 256×256×63 voxels, respectively.
Meanwhile, we had added background standard deviation (STD) and signal-to-noise ratio (SNR) to evaluate the performance for applying 2D and 3D MSO operation in Table 2 and Table 3 in this revision.
Table 2. The Deluxe Jaszczak phantom and a rat of background standard deviations were obtained by 2D-and 3D-MSO with percentage of reducing noise. The percentage of reducing noise is defined as (Background STD in Raw - Background STD after MSO)/(Background STD in Raw)×100%.
MSO matrix size |
Background STD in Phantom |
background STD in Rat |
Reducing Noise in Phantom |
Reducing Noise in Rat |
Raw1 |
750.59 |
6.27 |
||
2x2 |
238.94 |
4.91 |
68.2% |
21.7% |
2x3 |
174.48 |
4.64 |
76.8% |
26.0% |
3x2 |
171.67 |
4.65 |
77.1% |
25.8% |
3x3 |
110.64 |
4.19 |
85.3% |
33.2% |
3x3 ORC2 |
110.64 |
4.19 |
85.3% |
33.2% |
2x2x2 ORC |
141.15 |
3.65 |
81.2% |
41.8% |
2x3x2 ORC |
112.15 |
3.48 |
85.1% |
44.5% |
3x2x2 ORC |
104.50 |
3.50 |
86.1% |
44.2% |
3x3x2 ORC |
102.98 |
3.25 |
86.3% |
48.2% |
2x2x3 ORC |
112.53 |
3.29 |
85.0% |
47.5% |
2x3x3 ORC |
98.14 |
3.03 |
86.9% |
51.7% |
3x2x3 ORC |
98.20 |
3.24 |
86.9% |
48.3% |
3x3x3 ORC |
96.69 |
2.80 |
87.1% |
55.3% |
1 Raw data of FBP image without 2D-and 3D-MSO; 2 FBP image with 2D-MSO and ORC.
Table 3. The Deluxe Jaszczak phantom and rat of SNR were obtained by 2D-and 3D-MSO with percentage of increasing quality. The percentage of increasing quality is defined as (SNR after MSO – SNR in Raw)/(SNR in Raw)×100%.
MSO matrix size |
SNR in Phantom |
SNR in Rat |
Increasing Quality in Phantom |
Increasing Quality in Rat |
Raw1 |
4.53 |
14.21 |
||
2x2 |
5.19 |
14.53 |
14.6% |
2.3% |
2x3 |
5.41 |
15 |
19.4% |
5.6% |
3x2 |
5.41 |
14.81 |
19.4% |
4.2% |
3x3 |
5.76 |
15.38 |
27.2% |
8.2% |
3x3 ORC2 |
5.76 |
15.38 |
27.2% |
8.2% |
2x2x2 ORC |
5.51 |
15.39 |
21.6% |
8.3% |
2x3x2 ORC |
5.68 |
15.84 |
25.4% |
11.5% |
3x2x2 ORC |
5.73 |
15.89 |
26.5% |
11.8% |
3x3x2 ORC |
5.78 |
16.51 |
27.6% |
16.2% |
2x2x3 ORC |
5.55 |
15.85 |
22.5% |
11.5% |
2x3x3 ORC |
5.69 |
16.08 |
25.6% |
13.2% |
3x2x3 ORC |
5.66 |
15.67 |
24.9% |
10.3% |
3x3x3 ORC |
5.8 |
15.91 |
28.0% |
12.0% |
1 Raw data of FBP image without 2D-and 3D-MSO; 2 FBP image with 2D-MSO and ORC.
- How do you estimate the percentage of noise reduction?
[Reply]: Thanks reviewer constructive comments and suggestions.
The noise reduction was evaluated by the formula.
The percentage of reducing noise is defined as (Background STD in Raw - Background STD after MSO)/(Background STD in Raw)×100%.
The percentage of increasing quality is defined as (SNR after MSO – SNR in Raw)/(SNR in Raw)×100%.
- Could you provide some exemplary images showing how your method reduces the artifact?
[Reply]: Thanks reviewer constructive comments and suggestions.
In Figure 10 had shown the two examples to demonstrate how the MSO denoise the FBP images with respectively to Deluxe Jaszczak phantom and a rat PET FBP image.
Figure 10. Demonstration the input images by MSO ORC operation on the FBP images for the Deluxe Jaszczak phantom (a) and a rat (e). A post-processing image by a 2D 3×3 MSO (b, f) and a 2D MSO with ORC (c, g). A post-processing image by a 3D 3×3×3 MSO with ORC (d, h). Notice that the (f) and (g) are almost the same after 2D MSO with or without ORC in this study.

Reviewer 2 Report
A paper is nicely done. Computational time is very well elaborated. However, there are some problems...
1. Finding optimal filter coefficients by calculating all combinations is inefficient and old method. Nowadays, machine learning is used to shorten the search for optimal parameters. You should at leas address or compare morphology with genetic, CNN, SVD, EMD or similar methods.
If you noticed, there are not so many references which are 2020-2021, which means that morphology is obsolete as main method. However, it is integrated (and not addressed) in many ANNs and other machine learning methods.
2. Section 2.4. presents that this research requires high computational power and it is time consuming. Which hardware did you used? What was the execution time for the entire loop?
3. Section 2.9. "In this study, a unified and specific computer was used for computation time calculation of MSO processing." Is this computer normal (usual) for medical (clinical) conditions?
4. You could comment also https://doi.org/10.3390/jimaging7030044.
5. "It is believed that this technique can be used in pre-clinical experiments (e.g., rat studies), in pharmacokinetic experiments, in low-activity tracer imaging applications, and for patients who cannot tolerate long-duration tomography. " (Section 4) - Who believe? Authors or some reference?
6. A paper contains no proof that medical information is not changed. You should consult and survey several interdependent medical expert about this.
I would like to see complete research with survey of medical experts published.
Author Response
Reviewer 2
A paper is nicely done. Computational time is very well elaborated. However, there are some problems...
1. Finding optimal filter coefficients by calculating all combinations is inefficient and old method. Nowadays, machine learning is used to shorten the search for optimal parameters. You should at least address or compare morphology with genetic, CNN, SVD, EMD or similar methods. If you noticed, there are not so many references which are 2020-2021, which means that morphology is obsolete as main method. However, it is integrated (and not addressed) in many ANNs and other machine learning methods.
[Reply]: Thanks reviewer constructive comments and suggestions.
In this revision, the Table 5 was added in for comparisons the recently published literatures as below. Also, the following paragraph was added in the Section of Dissuasion.
We had added the comparisons among published literatures with the presented methods. Recently, the deep learning methods are popularly applied to medical and molecular images to reduce noise [24, 25] and enhance quality of images [26-33] including deep learning methods and generative adversarial networks algorithms (GAN). Mean-while, the classical methods were including patch-based regularization algorithms (PBRA) [25], Bayesian penalized likelihood reconstruction algorithm (BSREMA) [26], and 2D and 3D MSO with ORC were applied to perform noise reduction with acceptable results.
Table 5 shows the comparisons among published literatures with the presented methods.
Authors |
Year |
Modality |
Task |
Method |
Finding |
Gao et. al. [24, 25] |
2020 |
PET |
Reduce noise |
PBRA |
It is difficult to reduce noise and keep or improve quality of images due to encounter the low SNR of images or low count rates |
Seo et. al. [26] |
2020 |
PET |
Improvement quality |
BSREMA |
To enhances the quality of images and accuracy of qualification as per occurred in low-count rates of PET scanning |
Tatsumi et. al. [27] |
2021 |
PET/CT |
Improvement quality of image |
BPLRA |
To promote the gray levels of image as per low count rates condition |
Leuschner et. al. [28] |
2021 |
CT |
Improvement quality of image |
Deep learning methods |
The experimental results were shown and demonstrated to be able to improve the quality of noise images |
Yu et. al. [29] |
2020 |
Medical Image Synthesis |
Reduce noise, enhance quality |
Deep learning methods, 3D GAN |
To promote the quality of image under low count rates or low dose |
Soren et. al. [30] |
2020 |
CT. MRI, PET/MRI, PET/CT, |
Reduce noise, enhance quality |
GAN (GANs) |
These novel models made a great impact on the computer vision field |
Podgorsak et. al. [31] |
2021 |
CT |
CT artifact correction |
GANs |
Improvement of reconstructed image quality under sparse angles |
Koshino et. al. [32] |
2021 |
Medical and molecular imaging |
Reduce noise, enhance quality |
GANs |
GANs are promising tools for medical and molecular imaging for promoted quality of images. |
Wang et. al. [33] |
2018 |
PET |
Reduce noise, enhance quality |
3D GANs |
GANs are promising tools for improvement quality of image under Low count rates |
The Presented Methods |
2021 |
PET |
Reduce noise, enhance quality |
2D and 3D MSO with ORC |
Demonstrated efficiently perform denoising processing and obtained acceptable quality of images |
Notice: PBRA is Patch-based regularization algorithms; BSREMA is Block sequential regularized expectation maximization algorithm; BPLRA is Bayesian penalized likelihood reconstruction algorithm; Generative adversarial networks (GANs)
Section 2.4. presents that this research requires high computational power and it is time consuming. Which hardware did you used? What was the execution time for the entire loop?
[Reply]: Thanks reviewer constructive comments and suggestions.
Both the computing configurations and execution times were added in Table 1 and 4 in this revision.
Table 1. The computing configurations for calculating MSO operation.
Item |
contents |
Operating System |
Windows 64-bit operating system |
Central Processing Unit |
Intel Core 2 Quad CPU Q6600 |
Computer memory |
2.00 GB |
Table 4. The computational time in seconds by 2 and 3D MSO operation.
MSO matrix size |
Processing time(s) |
|
Deluxe Jaszczak phantom |
Rat |
|
2x2 |
37.86 |
53.73 |
2x3 |
184.08 |
288.88 |
3x2 |
184.29 |
287.39 |
3x3 |
1925.38 |
3445.9 |
3x3 ORC1 |
603.81 |
953.08 |
2x2x2 ORC |
26.38 |
25.83 |
2x3x2 ORC |
33.57 |
36.95 |
3x2x2 ORC |
64.36 |
36.89 |
3x3x2 ORC |
40.47 |
172.97 |
2x2x3 ORC |
34.26 |
80.1 |
2x3x3 ORC |
70.87 |
91.28 |
3x2x3 ORC |
36.66 |
91.24 |
3x3x3 ORC |
219.09 |
100.08 |
1 FBP image with 2D-MSO and ORC
Section 2.9. "In this study, a unified and specific computer was used for computation time calculation of MSO processing." Is this computer normal (usual) for medical (clinical) conditions?
[Reply]: Thanks reviewer constructive comments and suggestions.
The computer is a normal PC as shown in Table 1.
Table 1. The computing configurations for calculating MSO operation.
Item |
contents |
Operating System |
Windows 64-bit operating system |
Central Processing Unit |
Intel Core 2 Quad CPU Q6600 |
Computer memory |
2.00 GB |
- You could comment also https://doi.org/10.3390/jimaging7030044.
[Reply]: Thanks reviewer constructive comments and suggestions.
We had added in the above reference and also added more references about deep learning and other methods application for noise reduction and enhancement of image quality in this revision.
"It is believed that this technique can be used in pre-clinical experiments (e.g., rat studies), in pharmacokinetic experiments, in low-activity tracer imaging applications, and for patients who cannot tolerate long-duration tomography. " (Section 4) - Who believe? Authors or some reference?
[Reply]: Thanks reviewer constructive comments and suggestions.
We had omitted the original statements and modified the statements in this revision as below.
“The 2D and 3D MSO operation might be useful in the reduction of strike artifact in the FBP images. “
A paper contains no proof that medical information is not changed. You should consult and survey several interdependent medical experts about this.
[Reply]: Thanks reviewer constructive comments and suggestions.
We had omitted those of no proof medical information in this revision. Also, this revision was proven by the one of co-author, MD/Dr. Huei-Yong Chen.

Round 2
Reviewer 1 Report
All my comments from the previous revision have been addressed. I believe the manuscript has been sufficiently improved.
Author Response
Reviewer 1
All my comments from the previous revision have been addressed. I believe the manuscript has been sufficiently improved.
[Reply]: We thanks the reviewer constructive comments and encouragement in this revision.
Reviewer 2 Report
You have made a lot of efforts to comply to reviewers. Thanks for that. However, there are some problems remaining:
1. You cannot skip medical information when you talk about patients and PET. You should only have independent radiologist(s) working with PET to confirm your results obtained by the co-author. When I was working on medical diagnostics, ILO demanded that I have three independent radiologists.
2. I don't see scientific contribution in FBP or morphology. If you have planned to have original scientific contribution in image processing area, you should elaborate you contributions. Not just different application of known facts.
Author Response
Reply to Reviewer 2
You have made a lot of efforts to comply to reviewers. Thanks for that. However, there are some problems remaining:
1. You cannot skip medical information when you talk about patients and PET. You should only have independent radiologist(s) working with PET to confirm your results obtained by the co-author. When I was working on medical diagnostics, ILO demanded that I have three independent radiologists.
[Reply]: We thanks the reviewer constructive comments and encouragement in this revision.
It is very important to confirm medical information loss or not. The authors misunderstood the reviewer’s opinions in the previous round.
Thank a lot for your kindly reminding again.
- The rat images after MSO processing were check and exam by three senior independent radiologists in hospital already. The rat images after processing by MSO still retained the medical information including intensity in rat images and FWHM in phantom images.
- The estimated FWHM of six cold spots in phantom were closed to the designed diameters and shown in Figure 7 and Table 4 in this revision.
- In Table 3, the CR was added and evacuated the percentage of contrast as compared with raw FBP and after processing by MSO in this revision.
- I don't see scientific contribution in FBP or morphology. If you have planned to have original scientific contribution in image processing area, you should elaborate your contributions. Not just different application of known facts.
[Reply]: We thanks the reviewer constructive comments and encouragement in this revision.
The main contributions of the presented FBP+MSO are statements below and written in the Section of Conclusion in this revision.
(1). The contrast after processing by MSO was enhanced and less noise of the object in the FBP images.
(2). Efficiently reducing the background noise in FBP images.
(3). The geometrical FWHM estimated by MSO is almost the same as original FBP image by investigated cold spots in the phantom.

Round 3
Reviewer 2 Report
Changes are acceptable. How did you choose this journal? I think that there are better-suited journals for this paper.